# Opinion Dynamics Systems on Barabási–Albert Networks: Biswas–Chatterjee–Sen Model

**DOI:** 10.3390/e25020183

**Published:** 2023-01-17

**Authors:** David S. M. Alencar, Tayroni F. A. Alves, Gladstone A. Alves, Antonio Macedo-Filho, Ronan S. Ferreira, F. Welington S. Lima, Joao A. Plascak

**Affiliations:** 1Dietrich Stauffer Computational Physics Lab, Departamento de Física, Universidade Federal do Piauí, Teresina 64049-550, PI, Brazil; 2Departamento de Física, Universidade Estadual do Piauí, Teresina 64002-150, PI, Brazil; 3Departamento de Ciências Exatas e Aplicadas, Universidade Federal de Ouro Preto, João Monlevade 35931-008, MG, Brazil; 4Departamento de Física, Centro de Ciências Exatas e da Natureza, CCEN, Universidade Federal da Paraíba, Cidade Universitária, João Pessoa 58051-970, PB, Brazil; 5Departamento de Física, Universidade Federal de Minas Gerais, C. P. 702, Belo Horizonte 30123-970, MG, Brazil; 6Department of Physics and Astronomy, University of Georgia, Athens, GA 30602, USA

**Keywords:** opinion dynamics systems, Biswas–Chatterjee–Sen model, finite-size-scaling hypothesis, universality class, second-order phase transitions

## Abstract

A discrete version of opinion dynamics systems, based on the Biswas–Chatterjee–Sen (BChS) model, has been studied on Barabási–Albert networks (BANs). In this model, depending on a pre-defined noise parameter, the mutual affinities can assign either positive or negative values. By employing extensive computer simulations with Monte Carlo algorithms, allied with finite-size scaling hypothesis, second-order phase transitions have been observed. The corresponding critical noise and the usual ratios of the critical exponents have been computed, in the thermodynamic limit, as a function of the average connectivity. The effective dimension of the system, defined through a hyper-scaling relation, is close to one, and it turns out to be connectivity-independent. The results also indicate that the discrete BChS model has a similar behavior on *directed* Barabási–Albert networks (DBANs), as well as on Erdös–Rènyi random graphs (ERRGs) and *directed* ERRGs random graphs (DERRGs). However, unlike the model on ERRGs and DERRGs, which has the same critical behavior for the average connectivity going to infinity, the model on BANs is in a different universality class to its DBANs counterpart in the whole range of the studied connectivities.

## 1. Introduction

The interest in sociophysics has greatly increased in the past two decades, mainly when considering the dynamics that are present in social systems or networks [1,2,3,4,5,6,7,8]. Stauffer [5] and Galam [9], who proposed models that use local majority rule arguments, are considered the predecessors of the sociophysics emerging field, also termed as “the dynamics of opinions”. In fact, the dynamics of opinions are treated in the same way that researchers treat the usual real world [10]. As a result, opinion-dynamics systems have turned out to be one of the most studied subjects in sociophysics [7,11,12,13,14].

Biswas, Chatterjee, and Sen [8] proposed a continuous opinion dynamics model that is nowadays called the BChS model. This non-equilibrium system has pair interactions that can be either positive or negative, modeled by a single noise parameter *q*, that represents the fraction of negative interactions. The continuous version, on fully connected networks, has been studied through numerical simulations. It has been shown that a continuous phase transition takes place at a critical value $q=q_{c}$ with mean-field critical exponents [8]. On the other hand, this same model has been treated on regular lattices in two and three dimensions by using Monte Carlo (MC) simulations [15]. Although a continuous phase transition is also obtained, the criticality is now the same as the Ising model in the same dimensions.

The BChS model can also be defined on a discrete version and it has been studied by using Monte Carlo simulations on several scale-free networks and random graphs. The discrete model undergoes a second-order phase transition and, from the results that have been reported in the literature, we can say that: (*i*) its critical behavior is the same as the continuous version on fully connected networks and on regular two- and three-dimensional lattices [8,15]; (*ii*) on Apollonian networks (ANs), the critical behavior is different from the Ising model on regular lattices [16]; (*iii*) on directed Barabási–Albert networks (DBANs), the critical exponent ratios change with the connectivity of the networks [17], but the model belongs to the same universality class as the majority vote model (MVM) [18] on the same DBANs; (*iv*) on Erdös–Rènyi random graphs (ERRGs) and *directed* ERRGs random graphs (DERRGs) the critical exponents are different from the Ising model and also the MVM model on the same random graphs [19]. For additional details on the related MVM see, for instance, Refs. [20,21,22,23]

It is then well understood that it is not possible to know, a priori, whether a given model, on a particular scale-free network, will present a phase transition and, if a second-order phase transition occurs, what will be its universality class. For this reason, it will then be quite interesting to understand what should be the behavior of the discrete BChS model, when defined on BANs and, at the same time, compare the new behavior with the previous results of the model on other networks (and also with the MVM). Moreover, except for the Voronoi–Delaunay random lattices, the BANs seem to be the lacking most common scale-free topology to be studied within the context of the discrete BChS model [24]. In fact, as we will see below, there are some quite different and unexpected critical behaviors of the discrete BChS model on BANs, when compared to the DBANs and also the same model on other random network topologies.

Thus, in this work, the social consensus formation in the non-equilibrium discrete version of the Biswas–Chatterjee–Sen dynamic system on BANs, for several values of the noise parameter *q*, has been studied through Monte Carlo simulations. The scope of the paper is as follows. The next section presents the model, the MC simulations that have been employed, together with the evolution of the physical quantities from which the transition has been characterized. In Section 3, the obtained results are discussed and, in the last section, conclusions and some final remarks are addressed.

## 2. Model and Simulation

In order to study the BChS model [8,16,17,19] defined on Barabási–Albert networks, we will closely follow the procedure outlined in Ref. [19]. However, for questions of completeness, the model and simulations will be shortly summarized below.

Agents (or individuals) are set on each *i* node of a BAN with *N* sites. At time step *t*, the agents have opinion variables $o_{i}\left(t\right)$, which can assume three different values say −1, 0, or +1. The rules for updating $o_{i}\left(t\right)$ according to the BChS model are as follows:(i)The initial configuration is constructed by randomly assigning one of the three opinion states for each site *i* of the BAN;(ii)A site *i* is then randomly select to be updated;(iii)One bound of the site *i* is also randomly selected and an affinity $\mu_{ij}$ is given for this bond (*j* is the corresponding site sharing the bond with site *i*). This affinity parameter is another discrete variable that assumes a value +1, but can be turned negative with a probability *q*. The parameter *q* acts, in fact, as an external noise, modeling local discordances;(iv)The opinion variable of both sites sharing the selected bond are now updated following the rules
(1)$$\begin{array}{ccc} o_{i}\left(t+1\right) & = & o_{i}\left(t\right)+\mu_{ij}o_{j}\left(t\right)\;, \end{array}$$
(2)$$\begin{array}{ccc} o_{j}\left(t+1\right) & = & o_{j}\left(t\right)+\mu_{ij}o_{i}\left(t\right)\;, \end{array}$$
where $o_{i}\left(t\right)$ and $o_{j}\left(t\right)$ are the opinion states at time *t*, while $o_{i}\left(t+1\right)$ and $o_{j}\left(t+1\right)$ are the updated opinion states of the two sites *i* and *j*, respectively;(v)When the opinion state is out of the interval [−1,+1], for example being larger than +1, it is automatically made equal to +1. The same happens when the opinion state is smaller than −1, when it is made equal to −1.

An order parameter *O* for this dynamical system can be defined by averaging the opinion variables $o_{i}\left(t\right)$ over all individuals, i.e.,
(3)$$O=\left|\sum_{i}^{N}o_{i}\right|/N,$$
where *t* is large enough for the system reaching the stationary state. In the thermodynamic limit (infinite size networks), there is a critical value $q=q_{c}$, where for $q<q_{c}$ an ordered phase with O≠0 is present, while for $q>q_{c}$ one has a disordered phase with O=0 instead. Exactly at $q=q_{c}$, both ordered and disordered phases become equal at a second-order phase transition. It is interesting that, in this case, rather than a thermal driven critical behavior as in usual magnetic systems, one has indeed a kind of a random configuration driven phase transition.

This dynamic order–disorder transition can be studied, via MC simulations, using similar magnetic-like variables, so common at second-order phase transitions. For instance, for a given value of *q*, Equation (Equation 3) gives us a natural order parameter O(q). The fluctuation of the order parameter, $O_{f}\left(q\right)$, is in this way the analogous to the magnetic susceptibility. We can even define the equivalent of the reduced Binder cumulant of the order parameter, namely $O_{4}\left(q\right)$. These quantities are easily computed from MC simulations as
(4)$$\begin{array}{ccc} O(q) & = & [{\langle O\rangle }_{t}{]}_{av}, \end{array}$$
(5)$$\begin{array}{ccc} \;\;\;\;\;\;O_{f}\left(q\right) & = & N[{\langle O^{2}\rangle }_{t}-{\langle O\rangle }_{t}^{2}{]}_{av}, \end{array}$$
(6)$$\begin{array}{ccc} \;\;\;\;O_{4}\left(q\right) & = & 1-[\frac{{\langle O^{4}\rangle }_{t}}{3{\langle O^{2}\rangle }_{t}^{2}}{]}_{av}, \end{array}$$
where ${\langle \ldots \rangle }_{t}$ stands for time averages, that are computed after the system has reached the stationary state, and ${\langle \ldots \rangle }_{av}$ means the averages over different initial configurations. Some details of the simulations are described below.

In this work, the above quantities, as a function of *q*, have been computed through extensive MC simulations on BANs of finite sizes ranging from N=250, 500, 1000, 2000, 4000, 8000, up to 16,000. In order to let the system reach its stationary state, the initial $10^{5}$ MC steps (MCS) have been discarded. The corresponding time averages have then be computed by taking the next $2\times 10^{5}$ MCS. Here, one MCS consists of randomly choosing *N* sites of the network, as described above. For each set of network size *N* and parameter *q*, $10^{3}$ to $10^{4}$ different configurations have been considered to obtain the configurational averages.

Now, close to the critical noise parameter $q=q_{c}$, and for large system sizes, quantities (Equation 4), (Equation 5) and (Equation 6) should have a power-law behavior of the form (for further details see, for instance, Ref. [25])
(7)$$\begin{array}{ccc} \;\;\;\;\;O(q) & = & N^{-\beta /\nu }f_{O}\left(x\right), \end{array}$$
(8)$$\begin{array}{ccc} \;\;\;O_{f}\left(q\right) & = & N^{\gamma /\nu }f_{O_{f}}\left(x\right), \end{array}$$
(9)$$\begin{array}{ccc} \;O_{4}\left(q\right) & = & f_{O_{4}}\left(x\right), \end{array}$$
(10)$$\begin{array}{ccc} \;\;\;q_{c}\left(N\right) & = & q_{c}+cN^{-1/\nu }, \end{array}$$
where *c* is a non-universal constant and $q_{c}$ the corresponding critical noise for an infinite size graph. The critical exponents β, ν, and γ come from the behavior of the order parameter, correlation length, and fluctuation of the order parameter, respectively, when taken as a function of the noise parameter *q* close to $q_{c}$. In the above equations, however, we have the size behavior instead, where $f_{k}\left(x\right)$, with $x=N^{1/\nu }\left(q-q_{c}\right)$ and $k=\{O,\;O_{f},\;O_{4}\}$, are scaling functions. As we shall see below, for each finite graph of *N* sites, we can estimate its pseudo-critical noise parameter $q_{c}\left(N\right)$ by looking, e.q., at the peak presented by the fluctuation of the order parameter as a function of *q*.

We can now follow the standard procedure to obtain, numerically, the phase transition properties of the model in the thermodynamic limit. For a second-order phase transition, which will be the present case, the reduced Binder cumulant $O_{4}$, in Equation (Equation 9), should be independent of the system sizes for |x|<<1 (in fact, for large systems and close to $q_{c}$). This means that the crossing points of $O_{4}$, as a function of *q* and for different values of *N*, will give a good estimate of the critical noise parameter $q_{c}$ [25]. With $q_{c}$ in hands, and using Equations (Equation 7) and (Equation 8), one can compute the exponents ratio β/ν and γ/ν, respectively. Now, by locating the value of the noise parameter $q_{c}\left(N\right)$, where the maximum of $O_{f}$ occurs as a function of *q*, we can explore Equation (Equation 10) and further obtain an estimate of the critical exponent 1/ν.

Within this scheme, the main critical behavior of the system can be quantitatively specified. In addition, using the hyper-scaling hypothesis [26]
(11)$$2\beta /\nu +\gamma /\nu =D_{\mathrm{eff}}$$
it is also possible to compute the effective dimension $D_{\mathrm{eff}}$.

## 3. Results and Discussion

The dependence of the reduced Binder cumulant $O_{4}$ on the noise parameter *q*, for several finite sizes *N*, is shown in Figure 1a, for the smallest connectivity z=2, and in Figure 1b, for a quite higher value z=100. From these figures, one can see that: (*i*) the cumulant crossings attest indeed that we have a second-order phase transition in the system, where $q_{c}=0.173\left(1\right)$ for z=2 and $q_{c}=0.248\left(2\right)$ for z=100; (*ii*) from these values of $q_{c}$, we can see that the critical noise parameter is dependent on the connectivity *z*. Similar results are obtained for different connectivities and one can thus compute $q_{c}$ for several values *z*. Table 1 gives the results so obtained for other selected connectivities.

With $q_{c}$ in hand for different connectivities *z*, we can now compute the critical exponent ratios from the power-law relations of Equations (Equation 7), (Equation 8), and (Equation 10). The log–log plot of the quantities defined in Equations (Equation 4)–(Equation 6) and (Equation 10), as a function of *N*, on BANs with different connectivities *z*, are shown in Figure 2. All quantities have been computed at the extrapolated infinite network critical value $q_{c}$ for the corresponding connectivity *z*. The alignment of the data in those figures strongly corroborates the transition being indeed of second order, with the slope from a linear fit giving the desired critical exponent ratio. The critical exponent ratios so obtained are given in the legend of Figure 2 and also in Table 1, together with additional values of the connectivity *z*.

At this point, the universal behavior of the present finite-size-scaling functions, which are given in Equations (Equation 7)–(Equation 9), can be further tested in a still wider range of *q*. Note that we have, up to now, been working only with noises in the region close to the transition $q_{c}$. Figure 3 depicts the collapse of the present data of the scaled order parameter $ON^{\beta /\nu }$ (a), the scaled fluctuation of the order parameter $O_{f}N^{-\gamma /\nu }$ (b), and the scaled reduced Binder cumulant $O_{4}$ (c), as a function of the scaled displacement $\left(q-q_{c}\right)N^{1/\nu }$ for the particular connectivity z=20. We can see from the collapse of the data that we are not only dealing with a second-order phase transition, but also the computed critical exponents from the simulations seem to be indeed accurate.

The fluctuation of the order parameter $O_{f}\left(q\right)$ has a clear peak close to the connectivity transition, as is depicted in Figure 3b for z=20. The value of this peak $O_{f}\left(q_{c}^{max}\right)$, located at $q_{c}^{max}$ for each BANs size *N*, can also be used in Equation (Equation 8) to obtain an additional estimate of the critical exponent ratio γ/ν. Proceeding in this way, we obtain exponent ratios that are similar to those obtained from the same scaling relation using the previously computed $q_{c}$, for all values of *z*. This is clearly seen in Table 1 by comparing the numbers in the fifth and sixth columns, and also by inspecting Figure 2b,d.

Regarding the γ/ν quantity, it is worthwhile to stress that on DBANs [17], for all values of *z*, a different critical exponent ratio is obtained when considering the corresponding maximum value of the fluctuation of the order parameter $O_{f}\left(q_{c}^{max}\right)$. This can be seen in the fifth and sixth columns on Table 2, that reproduce the previous data of the same model on DBANs from Ref [17].

From Table 1 and Table 2, it is also possible to have a numerical comparison between the present results on BANs and those earlier obtained on DBANs, for several connectivities *z*. However, a global view of the data conveyed in Table 1 and Table 2 is better seen in the left panels of Figure 4. They plot the effective dimension $D_{eff}$, and the ratios β/ν and γ/ν, for the selected values of the connectivity *z*, on BANs (full circles) and on DBANs (open circles). From those plots we see that: (*i*) the effective dimension value $D_{eff}\approx 1.0$ for all *z* on both networks; (*ii*) after a slight variation for small values of *z*, the critical exponents become almost constant for z>20 on both networks; (*iii*) the critical exponent ratios on BANs are different from those on DBANs, meaning that they belong to different universal classes.

Although the effective dimension is close to one, the model has indeed a clear phase transition, in contrast to the same model on regular one-dimensional lattice, where no transition is observed. Nevertheless, we have to keep in mind that on these networks there are longer-range interactions, and not only short-range finite interactions that allows us to prove that one-dimensional classical models are free from any phase transition.

It is also interesting to compare the critical behavior of the discrete BChS model on BANs and DBANs to the critical behavior of the model previously studied on ERRGs and DERRGs [19] by using similar MC simulations. To make such comparison, we have the same quantities on ERRGs and DERRGs from Ref. [19] in the right panels of Figure 4. One can see that, on ERRGs and DERRGs, after an oscillatory behavior of the critical exponents for small values of *z*, the exponents are almost the same in both graphs, even in the limit of z→∞. This is due to the fact that in this limit, ERRGs and DERRGs have the same small world networks behavior, which does not happen to be the case on BANs and DBANs.

Finally, from the data of Table 1, one can notice that the value of the critical noise parameter increases with the connectivity, and approaches a limiting value as z→∞. This behavior is better seen in the phase diagram depicted in Figure 5 for the model on BANs. We also note, from Table 1 and Table 2, that $q_{c}$, for large values of *z*, is different for both networks. This is due to the fact that, unlike the ERRGs and DERRGs, in the limit z→∞ the BANs and DBANs do not have the same behavior as the small-world lattice. However, as for the model on ERRGs and DERRGs, the critical transition line does obey a similar inverse power law of the form
(12)$$q_{c}=A+B/z,$$
where *A* and *B* are non-universal constants. The inverse power-law of the transition line can be clearly seen in Figure 5b. From a linear fit to the data we obtain the constant values given in Figure 5b, from which we have the critical noise in the z→∞ limit as $q_{c}=0.2488\left(1\right)$. It is also useful to have Equation (Equation 12), since one can now estimate the critical noise $q_{c}$ for any value of *z*.

## 4. Concluding Remarks

The non-equilibrium BChS model, in its discrete version and defined on BANs, has been studied through Monte Carlo simulations for several values of the local consensus controlling parameter *q*. The results show that the model undergoes a second-order phase transition with critical exponents in a different universality class as the same model on regular two- and three-dimensional lattices [15], ERRGs and DERRGs [19], as well as on DBANs [17]. Despite of belonging to another universality class, the hyper-scaling [26] relation is always valid, independent of the connectivity *z*, and provides an effective dimension close to 1.0. It should be stressed that the critical exponents of the discrete BChS model on BANs are not the mean-field ones, as in the continuous version of the model on the same BANs [27].

Unlike the model on DBANs, the critical exponent on BANs, coming from the peak of the fluctuation of the order parameter, is comparable to the exponent directly obtained from the critical noise parameter. In addition, in the z→∞ limit, the results on BANs are different from those on DBANs, a result of the non-equivalence of both networks for large values of the connectivity. We believe that the present results on the BANs, in some sense complete the study of the discrete BChS model on scale-free networks and random graphs. Table 3 has a summary of the universality class of the BChS model on several networks and regular lattices.

## Figures and Tables

**Figure 1 entropy-25-00183-f001:**
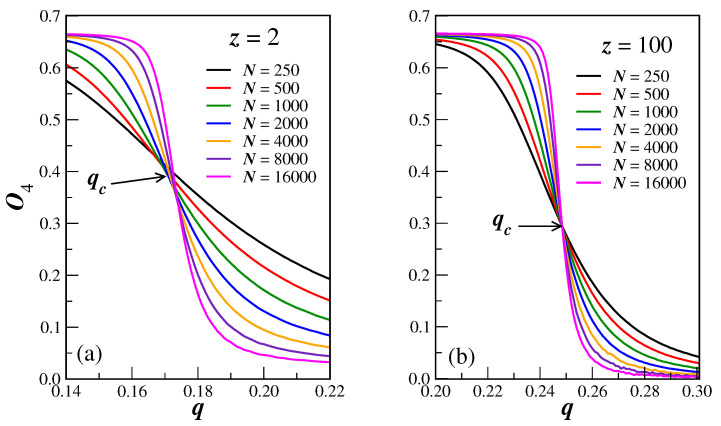
(color online) Reduced Binder cumulant $O_{4}$, plotted as function of the noise *q*, for connectivities z=2 in (**a**), and z=100 in (**b**). The sizes of the BANs *N* are specified in the legends. In these cases, the crossings occur, respectively, at $q_{c}=0.173\left(1\right)$ and $q_{c}=0.248\left(2\right)$.

**Figure 2 entropy-25-00183-f002:**
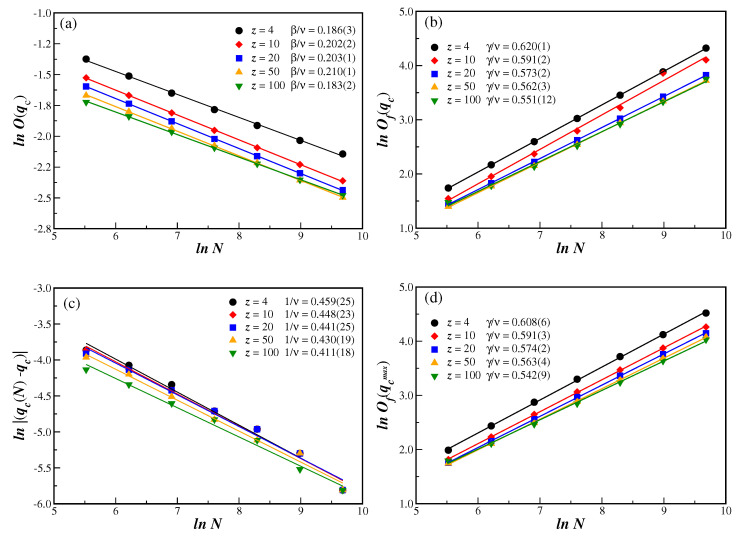
(color online) Log–log plot of the quantities defined in the text, as a function of *N*, on BANs with several values of the connectivity *z*. All quantities are computed at the extrapolated critical noise parameter $q_{c}$ for the corresponding connectivity *z*. (**a**) order parameter $O(q_{c})$; (**b**) order parameter fluctuation $O_{f}\left(q_{c}\right)$; (**c**) magnitude of the displacement $q_{c}\left(N\right)-q_{c}$; and (**d**) maximum amplitude $O_{f}\left(q_{c}^{max}\right)$. In all figures, the ratio of the critical exponents, that are obtained from the linear fits to the data (full lines), are given in the legends. The error bars are smaller than the symbol sizes.

**Figure 3 entropy-25-00183-f003:**
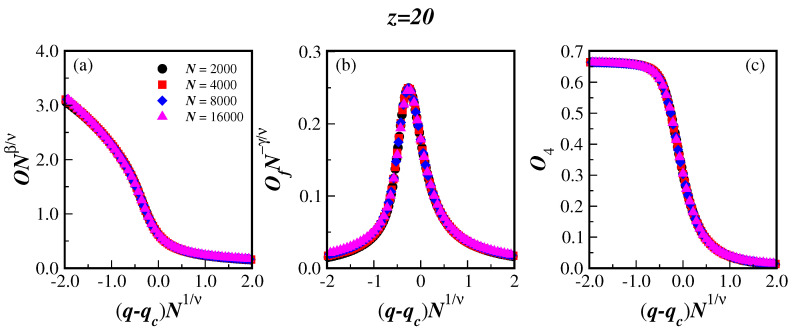
(color online) Collapse of the data of the scaled order parameter $ON^{\beta /\nu }$ (**a**), scaled fluctuation of the order parameter $O_{f}N^{-\gamma /\nu }$ (**b**), and scaled reduced Binder cumulant $O_{4}$ (**c**), as a function of the scaled displacement $\left(q-q_{c}\right)N^{1/\nu }$ for the connectivity z=20. The network sizes *N* are listed in the legend of panel (**a**) and also applies to panels (**b**,**c**).

**Figure 4 entropy-25-00183-f004:**
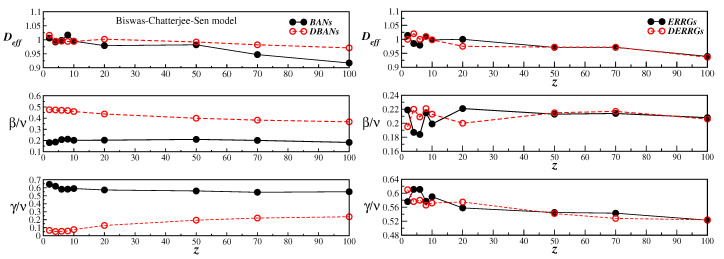
(color online) Effective dimension $D_{eff}$ (top panels), critical exponent ratios β/ν (middle panels) and γ/ν (bottom panels), as a function of the connectivity *z*. The left panels are the present results for the BChS model on BANs (full circles) and from Ref. [17] on DBANs (open circles). The right panels are, respectively, the results from Ref. [19] on ERRGs and DERRGs. The solid and dashed lines are, in all cases, only a guide to the eyes.

**Figure 5 entropy-25-00183-f005:**
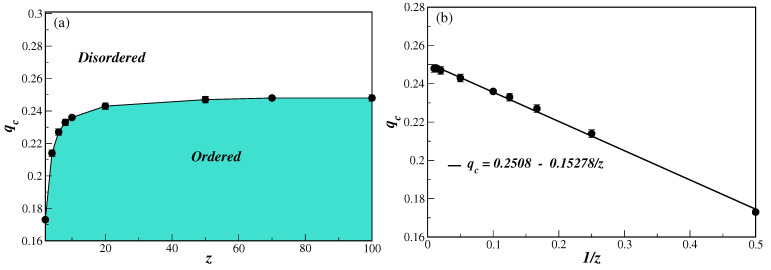
(color online) Phase diagram of the BChS model on BANs in the critical noise parameter $q_{c}$ versus connectivity *z* plane in (**a**), and its inverse 1/z in (**b**). The lines in (**a**) are just guide to the eyes, while in (**b**) they correspond to the best linear fit with the parameters given in the legend. In all cases, the error bars are smaller than the symbol sizes.

**Table 1 entropy-25-00183-t001:** Critical parameter $q_{c}$, critical exponent ratios 1/ν, β/ν, γ/ν, and effective dimension $D_{eff}$ of the BChS model on BANs. The selected values of the connectivity *z* are specified in the first column. Error bars are statistical only.

*z*	$q_{c}$	1/ν	β/ν	$\gamma /\nu (q_{c})$	$\gamma /\nu (q_{c}^{\max})$	$D_{\mathrm{eff}}$
2	0.173(1)	0.459(31)	0.181(4)	0.644(2)	0.638(9)	1.006(2)
4	0.214(1)	0.459(25)	0.186(3)	0.620(1)	0.608(6)	0.992(2)
6	0.227(2)	0.422(11)	0.208(2)	0.591(1)	0.599(4)	0.998(5)
8	0.233(2)	0.422(11)	0.212(1)	0.593(6)	0.595(3)	1.017(6)
10	0.236(1)	0.448(23)	0.202(1)	0.591(2)	0.591(3)	0.995(5)
20	0.243(2)	0.441(25)	0.203(1)	0.573(2)	0.574(2)	0.979(5)
50	0.247(1)	0.430(19)	0.210(1)	0.562(3)	0.563(4)	0.982(7)
70	0.248(1)	0.437(23)	0.201(3)	0.545(6)	0.551(5)	0.947(6)
100	0.248(2)	0.411(18)	0.183(1)	0.551(8)	0.542(9)	0.917(12)

**Table 2 entropy-25-00183-t002:** Results of the model on DBANs according to Ref. [17]. The legend of Table 1 also applies in this case, with the exception of the column regarding the exponent 1/ν, which has not been computed in [17].

*z*	$q_{c}$	β/ν	$\gamma /\nu (q_{c})$	$\gamma /\nu (q_{c}^{\max})$	$D_{\mathrm{eff}}$
2	0.439(3)	0.475(2)	0.066(1)	0.775(12)	1.016(4)
4	0.452(3)	0.473(1)	0.052(1)	0.829(10)	0.995(5)
6	0.448(3)	0.470(1)	0.055(2)	0.845(9)	0.995(6)
8	0.447(3)	0.468(1)	0.058(1)	0.860(9)	0.994(5)
10	0.443(3)	0.459(1)	0.076(1)	0.909(8)	0.994(3)
20	0.428(3)	0.437(1)	0.128(5)	0.890(10)	1.002(3)
50	0.408(2)	0.399(2)	0.194(6)	0.861(14)	0.992(9)
70	0.404(2)	0.382(2)	0.221(9)	0.850(2)	0.982(7)
100	0.388(4)	0.367(3)	0.237(8)	0.832(20)	0.971(31)

**Table 3 entropy-25-00183-t003:** Summary of the universality class of the discrete BChS model on several networks. All transitions are second order and the *class of its own* means that we are not aware yet of another member sharing the same critical exponents.

Discrete Biswas–Chatterjee–Sen Model		
network/lattice	universality	reference
fully connected	mean field	[8]
regular *d*-dimensional	*d*-dimensional Ising	[15]
Apollonian	class of its own	[16]
Barabási–Albert	class of its own *z* dependent exponents	this work
directed Barabási–Albert	majority vote model *z* dependent exponents	[18]
Erdös–Rènyi	class of its own *z* dependent exponents	[19]
directed Erdös–Rènyi	class of its own *z* dependent exponents	raquel
small world	z→∞ either of Erdös–Rènyi graphs	[19]
*Continuum Biswas–Chatterjee–Sen model*		
fully connected	mean field	[8]
regular *d*-dimensional	*d*-dimensional Ising	[15]

## Data Availability

Not applicable.

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
