# Peer review of "Opinion Dynamics Systems on Barabási–Albert Networks: Biswas–Chatterjee–Sen Model"

_entropy, 2023, doi:10.3390/e25020183_

Round 1
Reviewer 1 Report
The authors study a discrete opinion formation model in the BA network. They have found, with extensive numerical simulations and finite size analysis, the effective dimension (close to 1) and the universality (and its deviation) with the directed network version of the model.
It is very useful to study the dynamics of such opinion formation models in realistic networks. The actually connectivity of human interactions, which is often far from any idealized lattice structure, is a very relevant factor. Indeed, it is seen that the universality differs for directed and undirected versions of the network. Therefore, I recommend the manuscript for publication.
The authors should consider the following points for possible improvement of the presentation:
1. It is very curious to see that the effective dimension is one, or even less than that. Usually the model does not show phase transition when simulated in the one dimensional lattice. The authors should comment on that.
2. Along the same line, it is of interest to see what happens when a fraction of interactions allow for "extreme switches" i.e. -1 from +1 and vice versa. This can be done by changing the range of \mu. Even in mean field, the model does not have a finite critical point (p_c goes to 0) when the interactions are extreme. But given that the authors indicate that the model has a more robust behavior in the network form, including a fraction of extreme switches would be of interest.
3. I did not understand why there is a column of 1/nu in table 3, if it is to be kept fully empty. Is it not possible to get nu (or at least d\nu) from the data collapse of the Binder cumulant?
Author Response
“Summary of the changes made and response to the referees’
reports on Re: entropy-2133318”
Dear Ms. Ashely Kou
Assistant Editor
Entropy
Thank you for your email sending us the reports from the referees. We would also like to
thank the referees for their reports and for the constructive criticism on our manuscript.
In order to address the concerns raised by the referees, we have made some clarifications
in the body of the manuscript. We hope we have appropriately answered all the Referees’
questions.
We summarize below the changes we have made in the text, as well as a brief response
to all concerns or criticisms of the Referees. They are all written in red, both in the revised
manuscript and the text below.
Sincerely,
F. W. S. Lima and J. A. Plascak (on behalf of all authors)
I.
REPORT OF THE REFEREE 1
The authors study a discrete opinion formation model in the BA network. They have
found, with extensive numerical simulations and finite size analysis, the effective dimension
(close to 1) and the universality (and its deviation) with the directed network version of the
model.
It is very useful to study the dynamics of such opinion formation models in realistic
networks. The actually connectivity of human interactions, which is often far from any
idealized lattice structure, is a very relevant factor. Indeed, it is seen that the universality
differs for directed and undirected versions of the network. Therefore, I recommend the
manuscript for publication.
The authors should consider the following points for possible improvement of the presen-
tation:
11. It is very curious to see that the effective dimension is one, or even less than that.
Usually the model does not show phase transition when simulated in the one dimensional
lattice. The authors should comment on that.
This is indeed a good point. While classical models with finite short-range interactions
have no phase transition, the present model has a second-order phase transition with effective
dimension close to one. We can ascribe the transition to the fact that this is an effective
dimension, but the network has indeed longer-ranged interactions to allow the presence of
a phase transition. We have made some comments on this point on page ** of the new
manuscript.
2. Along the same line, it is of interest to see what happens when a fraction of interactions
allow for ”extreme switches” i.e. -1 from +1 and vice versa. This can be done by changing
the range of μ. Even in mean field, the model does not have a finite critical point (p c goes
to 0) when the interactions are extreme. But given that the authors indicate that the model
has a more robust behavior in the network form, including a fraction of extreme switches
would be of interest.
This is a very nice and pertinent suggestion. However, even with the fortran codes being
parallelized at the moment, we feel we do not have enough time to include such study in
this issue. However, this will surely be taken in to consideration in a future work.
3. I did not understand why there is a column of 1/nu in table 3, if it is to be kept
fully empty. Is it not possible to get ν (or at least dν) from the data collapse of the Binder
cumulant?
Thanks for pointing us out this detail. We have dropped this column in Table 2. And,
yes, it could be computed from the cumulant data, but this has not been, unfortunately,
done in (the new numbered) Ref. [19].
Have you also a Happy New Year!!! And thanks for the pertinent suggestions that surely
improved the text.
2

Reviewer 2 Report
This is a well-written manuscript that studies the Biswas-Chatterjee-Sen model in settings that were not considered before. Standard questions, that typically arise when scientists with physics background deal with social dynamics issues, such as characterization of phase transitions, are studied in this paper.
I cannot say that this manuscript offer somewhat brand-new. But the results outlined in the article are well-presented and clearly embedded in the existing body of literature on sociophysics. On this basis, I suppose that the article can be accepted. However, there are a few remarks that should be addressed before publication.
My other comments are very minor and referred to specific parts of the text:
1) Abstract. The usage of the abbreviation "BCS model" before its introduction.
2) Abstract. The whole sentence "The results also indicate that the discrete kinetic BCS model has a similar behavior as the model on directed Barabási-Albert networks (DBANs), as well as the same model on Erdös–Rènyi random graphs (ERRGs) and directed ERRGs random graphs (DERRGs)" is not clear. I do not understand what model do you mean when saying "... has a similar behavior as the model ..."
3) Intro. Sentence "Biswas, Chatterjee, and Sen [8] have proposed an interesting continuous opinion dynamics model that is nowadays shortly called BCS model."
I recommend discard the word "interesting". Note that I do not suppose that this model is not interesting. But the current version of the sentence is bit informal.
4) Intro. The sentence "... its critical behavior is the same as the continuum version ..." Continuous, I guess.
5) Section 2. Unfortunately, I am a reader who was not familiar with the BCS model before reviewing this article. While reading Intro, I noticed that there are at least 3 configurations of the BCS model: (i) continuous; (ii) discrete; (iii) discrete + kinetic. If I am correct, may I ask you to describe briefly all of them in Section 2? I believe that this will make the manuscript easier to read.
(As far as I understand, the current version of Section 2 presents the configuration (iii) of the BCS model.)
6) Section 2. "... Eqs. (4), (5) and (6) should have a power-law ..." Quantities (4), (5), and (6), I guess.
The following three comments are optional.
7) I would suggest a more complete review of the literature
8) I recommend to summarize your findings in a separate table. At the same place, I suggest to outline the previous results on the BCS model. Actually, you have already listed this information in the manuscript, for example: "... we can say that: (i) its critical behavior is the same as the continuum version on fully connected networks and on regular two- and three-dimensional lattices [8, 14]; (ii) on Apollonian networks ..." But I suppose that the table-based presentation would be more effective.
9) I recognize that formulas (7)-(11) are based on standard approaches from physics and are universally known and widely used by specialists in this field. However, I suppose, that the authors can describe their methods in more details.
Good luck and all the best in the coming New Year!
Author Response
“Summary of the changes made and response to the referees’
reports on Re: entropy-2133318”
Dear Ms. Ashely Kou
Assistant Editor
Entropy
Thank you for your email sending us the reports from the referees. We would also like to
thank the referees for their reports and for the constructive criticism on our manuscript.
In order to address the concerns raised by the referees, we have made some clarifications
in the body of the manuscript. We hope we have appropriately answered all the Referees’
questions.
We summarize below the changes we have made in the text, as well as a brief response
to all concerns or criticisms of the Referees. They are all written in red, both in the revised
manuscript and the text below.
Sincerely,
F. W. S. Lima and J. A. Plascak (on behalf of all authors)
I.
REPORT OF THE REFEREE 2
This is a well-written manuscript that studies the Biswas-Chatterjee-Sen model in settings
that were not considered before. Standard questions, that typically arise when scientists
with physics background deal with social dynamics issues, such as characterization of phase
transitions, are studied in this paper.
I cannot say that this manuscript offer somewhat brand-new. But the results outlined
in the article are well-presented and clearly embedded in the existing body of literature on
sociophysics. On this basis, I suppose that the article can be accepted. However, there are
a few remarks that should be addressed before publication.
My other comments are very minor and referred to specific parts of the text:
1) Abstract. The usage of the abbreviation ”BCS model” before its introduction.
1Corrected in the abstract
2) Abstract. The whole sentence ”The results also indicate that the discrete kinetic BCS
model has a similar behavior as the model on directed Barabsi-Albert networks (DBANs), as
well as the same model on ErdsRnyi random graphs (ERRGs) and directed ERRGs random
graphs (DERRGs)” is not clear. I do not understand what model do you mean when saying
”... has a similar behavior as the model ...”
Rereading this sentence we agree with the referee that it was rather confusing. We have
rewritten this sentence in the abstract hoping that now it is clearer.
3) Intro. Sentence ”Biswas, Chatterjee, and Sen [8] have proposed an interesting contin-
uous opinion dynamics model that is nowadays shortly called BCS model.”
I recommend discard the word ”interesting”. Note that I do not suppose that this model
is not interesting. But the current version of the sentence is bit informal.
Agreed and done.
4) Intro. The sentence ”...its critical behavior is the same as the continuum version ...”
Continuous, I guess.
Corrected.
5) Section 2. Unfortunately, I am a reader who was not familiar with the BCS model
before reviewing this article. While reading Intro, I noticed that there are at least 3 con-
figurations of the BCS model: (i) continuous; (ii) discrete; (iii) discrete + kinetic. If I am
correct, may I ask you to describe briefly all of them in Section 2? I believe that this will
make the manuscript easier to read.
(As far as I understand, the current version of Section 2 presents the configuration (iii)
of the BCS model.)
Sorry if we induced to some confusion regarding the configurations of the model. In fact,
this was due to the term kinetic, used as the dynamics of the model. In fact, we have only the
continuous and discrete version. Fortunately, you understood correctly the configurations.
We have, however, deleted the word kinetic all over the text in order not to induce the
readers to this same misinterpretation. We hope that now it is clearer.
6) Section 2. ”... Eqs. (4), (5) and (6) should have a power-law ...” Quantities (4), (5),
and (6), I guess.
You are right. Corrected!
The following three comments are optional.
27)I would suggest a more complete review of the literature.
We have tried to review the literature and in this way we have included new references
[12] and [20-23] (they are in red in the text).
8) I recommend to summarize your findings in a separate table. At the same place, I
suggest to outline the previous results on the BCS model. Actually, you have already listed
this information in the manuscript, for example: ”... we can say that: (i) its critical behavior
is the same as the continuum version on fully connected networks and on regular two- and
three-dimensional lattices [8, 14]; (ii) on Apollonian networks ...” But I suppose that the
table-based presentation would be more effective.
We liked very much the suggestion. We have then included a new table 3 in the manuscript
summaring the results.
9) I recognize that formulas (7)-(11) are based on standard approaches from physics and
are universally known and widely used by specialists in this field. However, I suppose, that
the authors can describe their methods in more details.
It is correct that these formulas are standard in the critical phenomena in general. How-
ever, we will have to quite extend the text to summarize how they are obtained using renormalization group ideas and finite-size-scaling hypothesis. Nevertheless, we have directed the
reader to reference [25], which gives a good idea on how these equations are obtained.
Good luck and all the best in the coming New Year!
Have you also a Happy New Year!!! And thanks for the pertinent suggestions that surely
improved the text.
3
